# Dispensing Antibiotics without a Prescription for Acute Cough Associated with Common Cold at Community Pharmacies in Shenyang, Northeastern China: A Cross-Sectional Study

**DOI:** 10.3390/antibiotics9040163

**Published:** 2020-04-06

**Authors:** Li Shi, Jie Chang, Xiaoxi Liu, Panpan Zhai, Shuchen Hu, Pengchao Li, Khezar Hayat, John Alimamy Kabba, Zhitong Feng, Caijun Yang, Minghuan Jiang, Mingyue Zhao, Hao Hu, Yu Fang

**Affiliations:** 1Department of Pharmacy Administration and Clinical Pharmacy, School of Pharmacy, Xi’an Jiaotong University, Xi’an 710061, China; shili1014@stu.xjtu.edu.cn (L.S.); emmazhai@stu.xjtu.edu.cn (P.Z.); hushuchen@stu.xjtu.edu.cn (S.H.); lipengchao1996@stu.xjtu.edu.cn (P.L.); khezar.hayat@uvas.edu.pk (K.H.); Kabbajohn@stu.xjtu.edu.cn (J.A.K.); fzt5051445@stu.xjtu.edu.cn (Z.F.); yangcj@mail.xjtu.edu.cn (C.Y.); jiangmh2017@xjtu.edu.cn (M.J.); mingyue0204@xjtu.edu.cn (M.Z.); 2Center for Drug Safety and Policy Research, Xi’an Jiaotong University, Xi’an 710061, China; 3Shaanxi Center for Health Reform and Development Research, Xi’an Jiaotong University, Xi’an 710061, China; 4School of Business Administration, Shenyang Pharmaceutical University, Shenyang 110016 China; 105000108@syphu.edu.cn; 5Institute of Pharmaceutical Sciences, University of Veterinary and Animal Sciences, Lahore 54000, Pakistan; 6State Key Laboratory of Quality Research in Chinese Medicine, Institute of Chinese Medical Sciences, University of Macau, Taipa 999078, Macao, China; HaoHu@um.edu.mo

**Keywords:** nonprescription antibiotic dispensing, pharmacy service practice, community pharmacy, acute cough, standardized client study

## Abstract

The inappropriate use of antibiotics is a major health issue in China. We aimed to assess nonprescription antibiotic dispensing and assess pharmacy service practice at community pharmacies in Shenyang, northeastern China, and to compare these practices between pediatric and adult cases. A cross-sectional study was performed from March to May 2018 using the standardized client method. Two different simulated scenarios were presented at pharmacies, namely, pediatric and adult acute cough associated with a common cold. Of 150 pharmacy visits, 147 visits were completed (pediatric case: 73, adult case: 74). A total of 130 (88.4%) community pharmacies dispensed antibiotics without a prescription, with a significant difference between pediatric and adult cases (pediatric case, 79.5% versus adult case, 97.3%, *p* = 0.005). Symptoms were asked in most visits (pediatric case: 82.2%, adult case 82.4%). Patients’ previous treatment and history of allergies were both inquired more frequently in the pediatric cases than in the adult cases. Medication advice was provided more often in the adult cases than in pediatric cases. Antibiotics were easily obtained without a prescription in Shenyang, especially for adult patients. Adequate inquiries and counseling had not occurred in most pharmacies.

## 1. Introduction

Antimicrobial resistance (AMR) has become a global health crisis [1]. In 2017, the World Health Organization (WHO) emphasized that AMR has severe effects on health and the economy, and the inappropriate use of antibiotics stimulated and accelerated the process of AMR [2].

Inappropriate use of antibiotics in China is a serious issue. China is one of the world’s largest producers and consumers of antibiotics. Antibiotics are widely used for disease treatment in humans and animals [3]. In 2013, 92,700 metric tons of antibiotics were consumed in China, 48% of which were consumed by humans, with the remaining consumed by animals. These usage estimates exceed the use in the UK and much of northern Europe [4]. Over the past decade, more than half of outpatients in China have been prescribed antibiotics—far above the level recommended by the WHO (less than 30%) [3].

In China, medical antibiotic use has been restricted to prescription-only since 2004 [5]. It is illegal to dispense antibiotics without a valid prescription of practicing physicians. The regulations have mandated the dispensing of antibiotics only under the guidance of a licensed pharmacist at community pharmacies [6,7]. Pharmacists have no role to prescribe, but they have the obligation to review the prescription before dispensing the antibiotics. However, the dispensing of antibiotics without a prescription at community pharmacies remains common [8].

The standardized client method, also known as the mystery shopper or simulated client method, is an internationally recognized tool for outcome measures in pharmacy practice research [9], and it has been used to assess the non-prescription sale of antibiotics at community pharmacies worldwide [10,11]. During a simulated visit, a standardized client presented as patients or relatives of patients, to approach pharmacy staff requesting medical assistance based on a predetermined scenario with a predefined standardized procedure [12]. Thereby, this method can provide an insight into the pharmacy staffs’ actual behavior of medicines’ dispensing and counseling [10,13].

Using standardized clients based on adult upper respiratory tract infections (URTIs) and pediatric diarrhea cases, our previous studies found that the prevalence of nonprescription sales of antibiotics is high in both urban and rural areas of different regions in China, and the quality of pharmacy services is inadequate [14,15]. However, knowledge regarding these issues in northeastern China is absent. Therefore, in this study, we used a standardized client method with hypothetical cases of acute cough to assess antibiotics’ dispensing practices and counseling services at community pharmacies in Shenyang, the important central city and the only megacity in northeastern China [16].

Secondly, our previous studies did not allow for a direct comparison between pharmacy practices responding to pediatric and adult patients based on the same clinical case. Children are more vulnerable to potential risks of medicines [17], and Chinese guidelines on clinical use of antibiotics explicitly detail the specific requirements of antibiotic use in children. According to the guidelines for clinical antibiotics use, aminoglycosides and glycopeptides antibiotics, both of which have ototoxicity and nephrotoxicity, should be avoided in children. With possible adverse reactions, tetracyclines antibiotics can not to be used for children under 8 years old, and quinolones should not be used for minors under 18 years old [18]. Thus, herein, we compared community pharmacy practices between pediatric and adult acute cough cases.

## 2. Results

We excluded two pharmacy visits due to the respondent staffs’ suspicion in each case and another one in the pediatric case because of the incomplete procedure. Finally, 73 visits based on the pediatric cases, and 74 visits based on the adult cases were completed. 

The characteristics of the community pharmacies surveyed and of the observed pharmacy staff members are reported in Table 1. After the multivariate binary logistic regressions, it was found that nonprescription antibiotics are less likely to be dispensed for a child with acute cough odds ratios (OR) = 0.099, 95% confidence interval (CI) 0.020–0.505, *p* = 0.005. Pharmacy type, size and location, the gender and age of staff were not significantly associated with the likelihood of the nonprescription antibiotics dispensing (Table 1).

### 2.1. Nonprescription Antibiotic and Other Medicines Dispensing

Overall, antibiotics were obtained without a prescription from 130 (88.4%, (95% CI 83.2–93.70)) of the 147 pharmacies. The proportions of dispensing antibiotics without a prescription were 22.5% (15.6–29.3), 60.5% (52.5–68.5), and 5.4% (1.7–9.2) in demand level 1, level 2, and level 3, respectively (Figure 1). All the pharmacy staff refused to dispense antibiotics on public health grounds and not on administrative reasons. As for non-antibiotics medicines, 28.6% were prescription medicines and 72.4% were over the counter medicines. 

#### 2.1.1. Medicine Dispensing in Pediatric Acute Cough

Non-prescription dispensing of antibiotics occurred in 58 (79.5%; (95% CI 70.0–88.99)) of the 73 pharmacy visits. Antibiotics were recommended initially in only 15 (20.6% (11.1–30.0)) interactions. Most antibiotic dispensing occurred following the simulated patients’ requests: 39 (53.4% (41.7–65.1)) interactions under the demand level 2, and 4 (5.5% (1.1–10.8)) interactions under the demand level 3 (Figure 1).

After excluding antibiotics recommended at the third level, the most frequently dispensed antibiotics were cephalosporin (35.8% (95% CI 25.1–46.5)), followed by azithromycin (29.6% (19.4–39.8)) and roxithromycin (16.1% (7.9–24.2)) (Figure 2) (Table A1).

More than one medicines were dispensed in the majority of pharmacies. Two medicines were dispensed in 53 (72.6% (95% CI 62.1–83.1)) visits and three medicines were dispensed in 15 (20.5% (11.1–30.0)) visits. Detailed information about the type of dispensed medicines in each visit shown in Table A2 in the Appendix A. Cough medicines were dispensed in 72 pharmacies (98.6%), cold medicines were dispensed in 11 pharmacies (15.1%), and Vitamin C was dispensed in 2 pharmacies (2.8%). Anti-inflammatory Chinese patent medicine was dispensed in 15 pharmacies (20.5%).

#### 2.1.2. Medicine Dispensing in Adult Acute Cough

Non-prescription dispensing of antibiotics occurred in 72 (79.5%; (97.3%; 95% CI 93.5–100.0)) of the 74 pharmacy visits. Antibiotics were also recommended initially in only 18 (24.3% (14.3–34.3)) interactions. Most antibiotic dispensing occurred following the simulated patients’ requests: 50 (67.6% (56.6–78.5)) interactions under the demand level 2, and 4 (5.4% (0.1–10.7)) interactions under the demand level 3 (Figure 1).

After excluding antibiotics recommended at the third level, the most frequently dispensed antibiotic was azithromycin (30.0% (95% CI 20.9–39.1)), followed by cephalosporin (28.0% (19.0–37.0)) and roxithromycin (26.0% (17.3–34.7)) (Figure 2) (Table A1).

Most pharmacies also dispensed more than one medicine for the adult. Two medicines were dispensed in 56 (75.7% (95% CI 65.7–85.7)) visits and three medicines were dispensed in 16 (21.6% (12.0–35.2)) visits. Cough medicines were dispensed in 72 pharmacies (97.3%) and cold medicines were dispensed in 9 pharmacies (12.2%). Anti-inflammatory Chinese patent medicine was dispensed in 15 pharmacies (16.0%) (Appendix A
Table A2).

### 2.2. Pharmacy Service Practice

#### 2.2.1. Inquiries and Counseling

As for the proportion of major inquiry and counseling, only the proportion of advice provision has a significant difference between the two cases (pediatric case 23.3% versus adult case 47.3%, *p* = 0.002) (Table 2). 

In both cases, symptoms were asked most frequently (pediatric: 82.2%, adult: 82.4%). The most frequently (pediatric: 74.0%, adult: 79.7%) asked question was, “is there sputum?”. Medical history was rarely asked (pediatric: 2.7%, adult: 4.0%). Previous treatment and history of allergies were inquired more frequently in the pediatric case (21.9%, 39.7%, respectively) than in the adult case (10.8%, 31.1% respectively). The most frequent advice provided was about usage (pediatric: 21.9%, adult: 40.5%). Only 4 (5.5%) pharmacies in the pediatric case recommended patients to seek medical treatment if the drug is ineffective. Among 29 pharmacies recommended cephalosporin in the adult case, only 3 pharmacies (10.3%) advised patients to stop alcohol while taking cephalosporin. All pharmacies provided advice by oral expression. The frequency of specific inquiries and counseling and chi-square test results shown in Table 2.

#### 2.2.2. Other Inquired Information

Before dispensing drugs, confirming who is the patient and the patient’s age occurred in 114 pharmacies. The requirement of the dosage form of drugs was inquired about in 14 pharmacies, the reserve of antibiotics at home was asked in 12 pharmacies, and the reserve of cold medicines at home was inquired about in 2 pharmacies. In 2 pharmacies, at the beginning of the visit, the staff directly inquired about the client’s product demand.

## 3. Discussion

This study showed that dispensing non-prescription antibiotics for patients with acute cough in Shenyang urban area was common. The most frequently dispensed class of antibiotics was Macrolides. Shenyang, the important central city in northeast China approved by the State Council, is also the only megacity in northeast China [16,19]. From the practice in Shenyang, dispensing non-prescription antibiotics may also be common in northeast China. Compared to our previous multi-center study in urban China [14], the level of inappropriate antibiotics dispensing in Shenyang is likely to be worse than that in Nanjing, Changsha, and Xi’an. Non-prescription antibiotics are more likely to be dispensed for adults. Most of the pharmacies (79.5%) in the pediatric case, while almost all pharmacies (97.3%) in an adult case, had dispensed antibiotics. Antibiotics were dispensed significantly less for children, possibly due to the increased risk of adverse events in children, and people pay more attention to children’s medication safety [18]. In this study, Quinolones, which are contraindicated for the children [17], were only dispensed in two pharmacies for the adults. The pharmacy service was also slightly better for the pediatric case, but medical advice was provided more often in the adult case, which may occur due to the fact that the staff thought children’s caregivers might also pay attention to the usage of the medicine [20].

An acute cough is an important circumstance for inappropriate antibiotic use. Acute cough is often associated with the common cold and acute tracheobronchitis, both of which are usually caused by a virus [21]. Symptomatic treatment is the principle to treat acute cough. Using antibiotics is unnecessary, will not shorten the duration of a cold or relieve symptoms and only with slight benefits for acute tracheobronchitis to shorten the duration of acute cough for about 16 hours, but may be accompanied by adverse reactions and antibiotic resistance [21,22,23,24,25]. When there is evidence of a bacterial infection, oral antibiotics such as β-lactams and quinolones can be used [21]. In our study, dispensing of inappropriate antibiotics such as azithromycin and roxithromycin also frequently occurred. In other simulated studies [10,26,27,28], they also found antibiotic dispensing for upper respiratory infections. Inappropriate antibiotic dispensing for URTIs is an important reason for antibiotic abuse. It is crucial to manage antibiotic use on upper respiratory infections. Meanwhile, a survey commissioned by the WHO about public awareness in China showed that 61% thought antibiotics were effective against common colds and flu [29]. It is crucial to improve public awareness on antibiotics usage for an upper respiratory infection.

Focusing on disease management rather than antibiotics might promote the rational use of medicines more effectively. In almost every pharmacy, we found that more than two medicines were dispensed together. Antibiotics, cough medicines, and cold medicines were frequently dispensed to the client together. This has undoubtedly increased unnecessary drug use, side effects, and consumer drug expenditure. The majority of dispensed medicines were Chinese traditional patent medicines. Chinese patent medicine is also commonly used in hospitals. A study conducted in rural Beijing showed that the percentage of encounters with traditional Chinese patent medicines prescribed was 52.5% in general practice clinics [30]. Another study found that when the use of antibiotics decreased following intervention targeting irrational antibiotic use, the use of traditional Chinese medicine increased, which suggests that the Chinese traditional patent medicines for antibiotics may be used as substitutes for antibiotics to reassure patients [31]. 

During the study, most antibiotics were dispensed under the client’s demand level 2 (request antibiotics). This result is in line with the previous finding that customer expectation is a contributing factor for non-prescription antibiotic dispensing [10,32]. In China, self-medication with antibiotics is prevalent. Some people think antibiotics are anti-inflammatory drugs and that is effective for cold and flu, and that antibiotics can shorten the duration of a cold or relieve symptoms [8,28,33,34]. Medication habits and inappropriate awareness increase the public expectation for antibiotics treatment. In the whole survey, no pharmacies refused to dispense antibiotics because of administrative reasons. Among other dispensed medicines, expect for antibiotics, a quarter of these were also prescription-only medicines but were also dispensed without requiring a prescription. Non-compliance with the regulations may be the primary reason to push non-prescription sales of antibiotics in pharmacies [10]. Pharmacy staffs’ knowledge and attitude are also decisive in antibiotic-dispensing practice. It is vital to strengthen law enforcement and improve pharmacy staffs’ knowledge about antibiotics through intervention and education. 

This study also showed that there was no ideal management for acute cough and no good pharmacy service. Acute cough may also be an early sign of serious disease and foreign body aspiration or be caused by the exacerbation of some illness and environment factors [21,22,23,24]. Community pharmacy staff should ask adequate questions from patients with a cough before dispensing medicines. Referring patients to the hospital or seeking specialist advice on further investigation occurred rarely. Most pharmacies just completed a partial inquiry of symptoms. Only a small portion (10.2%) of pharmacy visits involved questions about sputum color, which, to some extent, can indicate whether it is a bacterial or viral infection [20]. Inquiries, including the duration, the length of cough, and accompanying symptoms, which are used to determine whether the cough is acute or chronic, including the cause of the cough, were barely asked. About 20% of pharmacies did not ask anything further about the symptoms. Empirical drug treatment for cough may delay further diagnosis. Lack of a definitive diagnosis is also the driver for non-prescription dispensing of antibiotics [35]. At the same time, approximately one-third of the pharmacies asked about drug allergies history and recommended medications that cannot guarantee the safe and rational use of medication. Increasing pharmacists and pharmacy staffs’ access to clinical guidelines on respiratory infections might be beneficial for the health system.

Our findings highlight the urgency for policy makers to develop multifaceted approaches to the steward of antibiotic dispensing in retail settings. For the public, a multi-measures education intervention is necessary to improve public knowledge about upper respiratory infection and antibiotics use. In Italy, there was a national campaign which improved public awareness [36]. It will be meaningful if the government initiates regular campaigns against antibiotic abuse, such as having health educations for students, strengthening the construction of public health at the community level to popularize knowledge about related diseases and medicines. For pharmacies, national antibiotics stewardship in private pharmacies is necessary. In low-income and middle-income countries, public stewardship on private pharmacy practices showed that regulation and training could enhance private pharmacies’ adherence to recommended practice [37]. First, further reinforcement of regulations is needed. The government should strengthen the supervision and punishment for the sale of antibiotics without prescriptions. Establishing a surveillance system capable of tracking medicine sales at every community pharmacy could be an effective supervision measure. Second, regular training and on-the-job education for pharmacy staff about the regulation, the management of common illness, and appropriate use of antibiotics are essential. 

There were several limitations to our study. First, some observational indicators, such as the age range of the attending pharmacy staff, were perceived by one simulated client, so inevitably, it may be subject to inaccuracy. Second, in this study, we did not distinguish whether the respondent was a licensed pharmacist or pharmacy assistant. Therefore, we did not explore the difference in pharmacy practices between those provided by the pharmacists and pharmacy assistants. Third, we only performed one simulated case in every sampled pharmacy. Hence, we were unable to pair the observations of pediatric and adult cases for each pharmacy to enhance the analysis of differences in pharmacy practices in pediatric cases and adult cases. However, during the survey, the standardized client visited two pharmacies with the same geographical location and similar size pharmacies for one child case and one adult case. Fourth, since the observational items were collected by memorizing the information, the data collected might be subject to some extent of ‘recall bias’. However, in our study, the standardized client had done sufficient exercise to be competent in memorizing and recording all the needed items accurately. A standardized data collection form was used, and the data of the needed observational items in each pharmacy visit had been recorded in the form shortly after the visit. Moreover, one researcher, serving as the standardized client, conducted all the visits and therefore, the consistency of all the visits was high and might counteract with the drawback.

## 4. Methods

### 4.1. Study Setting and Pharmacy Selection

This cross-sectional study was carried out in Shenyang from March to May 2018, using a standardized client method by simulating pediatric and adult acute cough.

Shenyang is the capital city of Liaoning province, which is located in northeastern China. The city has 13 administrative divisions, including 5 districts in central urban areas, 4 districts in suburbs, and 4 counties. We surveyed community pharmacies in all the districts in central urban areas and suburbs. Four counties were excluded because of the low density of pharmacies and the distance from the downtown. All community pharmacies in the sampled districts were identified as eligible for inclusion in the study.

Convenience sampling was used to select pharmacies. In each selected district, every sub-district was numbered, and we first selected a sub-district as a central location in each district using random numbers generated in Excel 2013. Then, we randomly selected a pharmacy in the central location as a central pharmacy, and starting from the central pharmacy, the surveyed community pharmacies were selected at random in four distinct directions—north, west, south, and east—from those within a 1 h drive.

There was only one visit in each pharmacy. All pharmacy visits were performed in the order of “a pediatric case visit, an adult case visit, a pediatric case visit, an adult case visit”. Put another way, based on the survey, the odd-numbered visits were for pediatric cases, and the even-numbered visits were for adult cases.

### 4.2. Simulated Scenarios and Standardized Client

To compare the response of pharmacy staff to adults and children, two scenarios, acute cough associated with the common cold, of both patients, were developed. We chose acute cough associated with the common cold as a hypothetical case for the following reasons. Firstly, it is a minor illness of upper respiratory tract infections (URTIs) and easily reproducible for standardized clients. Having symptoms of a cough for less than 3 weeks can be defined as an acute cough [21,22,23,24]. Secondly, it occurs recurrently. Patients with URTIs are more likely to self-medicate with [26] and be dispensed antibiotics without prescriptions in community pharmacies [27]. We assumed that pharmacy staff are willing to dispense antibiotics to cough patients, the cough patients who also would like to seek drugs and advice through community pharmacies. Thirdly, acute cough associated with the common cold is usually caused by a virus [22]. Using antibiotics will not shorten the course of a cold or alleviate symptoms but may be accompanied by adverse reactions and antibiotic resistance [21,22,23,24]. Dispensing antibiotics for an acute cough is a case of misuse of antibiotics. Fourthly, the aggravation of asthma, chronic bronchitis, and bronchiectasis, and medication history of angiotensin-converting enzyme inhibitors can lead to an acute cough. Foreign body aspiration, smoking, and environmental factors are also the causes of acute cough. Because of the complex causes of an acute cough, to avoid the masking of underlying illness processes, cough remedies and medicines are not recommended to be dispensed to such types of patients without having complete medical information about their illness [21,22,23,24]. 

In the pediatric case, the patient was a 4-year-old boy. In the adult case, the patient was a 48-year-old man. The specific patient description, designed based on symptoms of acute cough caused by the common cold, included the duration and period of cough: cough for two weeks and cough during daytime, and other mild accompanying symptoms: a little white sputum, a sore throat, and a runny nose. No other medical and allergy history and no previous treatment for patients were given. The ideal management for acute cough was that patients were asked for specific symptoms and medical history by pharmacy staff before recommending drugs or recommending a referral if necessary, and no antibiotics were dispensed. 

Both cases were based on a “third person” scenario [12]. The simulated client acted as the aunt of the child in the pediatric case and the daughter of the man in the adult case. One of the researchers of this study (L.S.) was a simulated client, who is a female, 23-year-old undergraduate pharmacy student, familiar with simulation scenarios research design. The pharmacy visits were either of the pediatric case or the adult case. 

### 4.3. Procedures

To obtain the maximum standardization and to maintain the consistency of the simulations, the standardized visiting procedure and script of presentation Appendix A (Figure A1) were designed. Each pharmacy visit was carried out strictly according to the predefined procedure. To avoid the “Hawthorne Effect”, all community pharmacies were not informed before the visit. Given that we used the standardized client method, the committee permitted a waiver of informed consent from pharmacies. Ethics approval was obtained before the pilot study.Pilot visits, before field surveys, were conducted to confirm the feasibility of the study and to test the validity of the data collection form. Additionally, these visits helped in ensuring that the information collected was enough to reflect the issue we wanted to analyze.After pilot visits, several additional visits outside the formal sampling frame were taken for training purposes. Once the interaction with pharmacy staff in real community pharmacies was made, the simulated client built confidence and became familiar with the standardized simulated visit process.During the formal simulated visits in the survey, the simulated client was presented with acute cough associated with the common cold of either the pediatric or adult case in community pharmacies. Three levels of demand, (level 1: client required some medicine for cough) and (level 2: client explicitly expressed the requirement of antibiotics, and demand level 3: client specifically required roxithromycin), were designed to quantify antibiotic dispensing practices in community pharmacies. The major items to evaluate pharmacy service included inquiries about symptoms, medical history, previous treatment, allergies, and advice provision. After the visit, the standardized client found an excuse to leave without telling the pharmacy about her real identity. A standardized data collection form (Appendix B Data collection form), consisted of pharmacy demographics including location, type, and scale of pharmacy, pharmacy staff characteristics, drug dispensing practices, and planned inquiry and counseling items, was filled out within 15 minutes after leaving the pharmacy (out of sight of the pharmacy staff).

### 4.4. Data Analysis

The total sample size was 150 and was equally divided into two scenarios, 75 for the pediatric case and 75 for the adult case. The sample size in each urban district was determined according to the proportion of residents (Appendix A
Table A3).

The primary outcome is nonprescription antibiotic dispensing practice, including the proportion of non-prescription antibiotics in three demand levels, and the type of antibiotics under demand level 1 and 2. The secondary outcome is the proportion of inquiries about the symptoms, medical and allergic history, previous treatment, and advice provision. Additionally, we also reported the other medicines dispensed beside antibiotics.

Descriptive statistical analyses were reported as the proportions of dispensing antibiotics without a prescription, and the percentages of each item were questioned. The Chi-square test was used to compare the difference in pharmacy practices between the pediatric case and the adult case. A multivariate binary logistic regression was used to examine factors associated with the nonprescription antibiotic dispensing. Factors associated with the nonprescription antibiotic dispensing include pharmacy characteristics, pharmacy staff characteristics, and the simulated case. We calculated the proportion with a 95% confidence interval (CI) for our main outcomes. All statistical analyses were performed using Statistical Program for Social Sciences (SPSS) 13.0. A *p*-value < 0.05 was considered statistically significant.

### 4.5. Ethics

Our study protocol has received ethics approval from the Biomedical Ethics Committee for Medical Research of Xi’an Jiaotong University (Approval number 2018-515).

## 5. Conclusions

In conclusion, the vast majority of community pharmacies sell antibiotics to consumers without a prescription. An acute cough is an important condition for inappropriate antibiotic dispensing, and it is crucial to managing antibiotic use of URTIs. Pharmacy staff inadequately inquired about the patient’s medical conditions but were more concerned with adverse reactions in children in the process. The government needs to formulate and improve stringent policies and measures including education of the general public, sales management of community pharmacies, and the guidance of pharmacy practices, to further prevent and control the unjudicial use of antibiotics.

## Figures and Tables

**Figure 1 antibiotics-09-00163-f001:**
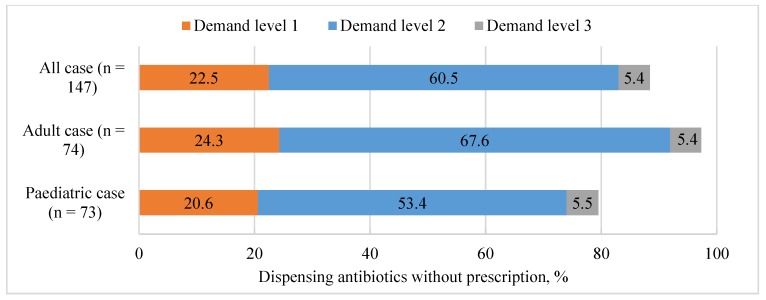
Proportions of dispensing antibiotics without a prescription in three different levels of demand. Demand level 1 (client required some medicine for cough), Demand level 2 (client explicitly expressed the requirement of antibiotic), Demand level 3 (client specifically required roxithromycin).

**Figure 2 antibiotics-09-00163-f002:**
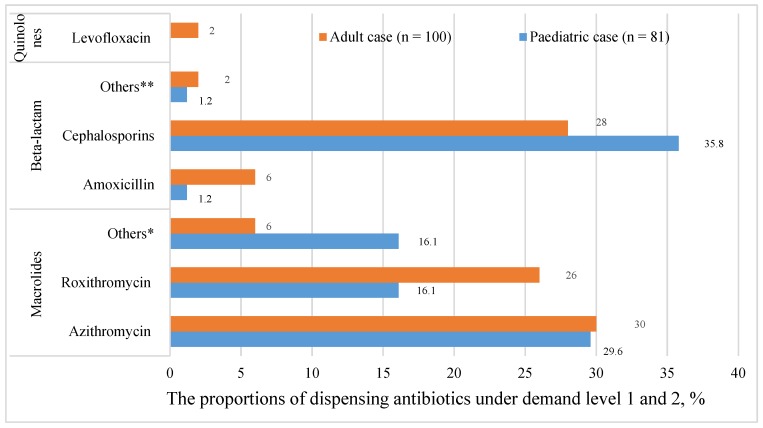
The proportions of dispensing antibiotics under demand levels 1 and 2. (n): the number of dispensed drugs; *: pediatric: cycloerythromycin (6), etoerythromycin (8); adult: roxithromycin ambroxol (5), erythromycin (1); **: pediatric: penicillin (1); adult: penicillin (1), amoxicillin clavulanate (1).

**Table 1 antibiotics-09-00163-t001:** Perceived characteristics of the visited pharmacies and respondent pharmacy staff and factors associated with non-prescription dispensing of antibiotics at community pharmacies in Shenyang.

Variables	Total Number	Nonprescription Antibiotic Dispensing		Multiple Logistic Regression		
	n(%)	Yes, n(%)	No n(%)	OR	95%CI	*p*-Value
**Case**						
Adult case	74(50.3)	72(97.3)	2(2.7)	ref		
Pediatric case	73(49.7)	58(79.5)	15(20.5)	0.099	0.020–0.505	0.005
**Pharmacy characteristics**						
**Socioeconomic** ^1^						
Low socioeconomic level	96(65.3)	85(88.3)	11(11.7)	ref		
High socioeconomic level	51(34.7)	45(88.2)	6(11.8)	0.814	0.225–2.943	0.753
**Type**						
Independent	18(12.2)	18(100)	0(0)	ref		
Chain	129(87.8)	112(86.8)	17(13.2)	0.000	0.000–	0.998
**Area**						
Suburb district	43(39.3)	40(93.0)	3(7.0)	ref		
Urban district	104(60.7)	90(86.5)	14(13.5)	0.892	0.204–3.914	0.881
**Distribution**						
Community center	112(76.2)	103(91.9)	9(8.1)	ref		
Medical center	17(11.6)	13(76.4)	4(23.6)	0.317	0.069–1.463	0.141
Shopping center	18(12.2)	14(77.8)	4(22.2)	0.370	0.078–1.765	0.212
**Size** ^2^						
Small (<100 m^2^)	68(46.3)	60(88.2)	8(11.8)	ref		
Medium (100–300 m^2^)	70(47.6)	61(87.1)	9(13.9)	1.479	0.446–4.906	0.522
Larger (≥300 m^2^)	9(6.1)	9(100)	0(0)	1.841E8	0.000–	0.999
**Staff characteristics**						
**Gender**						
Female	144(97.9)	127(88.2)	17(11.8)	ref		
Male	3(2.1)	3(100)	0(0)	8.938E8	0.000–	0.999
**Ages (years)**						
>50	6(4.1)	5(83.3)	1(16.7)	ref		
30–50	96(65.3)	86(89.6)	10(10.4)	10.613	0.313–359.761	0.189
≤30	45(30.6)	39(86.7)	6(13.3)	16.852	0.439–647.711	0.129

Note: The estimated odds ratios with 95% CI were obtained from the multivariate binary logistic regressions and all the independent variables with random intercepts were analyzed jointly in the regression model. ^1^ Socioeconomic: According to the gross domestic product (GDP) statistics, we categorize these districts with GDP higher than the average level into the high socioeconomic status region and the others into the low socioeconomic status region. Pharmacy characteristics of low/high socioeconomic level means a pharmacy in the low/high socioeconomic status region. ^2^ Size: According to the Good Supply Practice, the area of the community pharmacy’s place of business in the main urban district of the state (city) level shall not be less than 100 square meters. Our pharmacy visits were in major district of the city of Shenyang. Therefore, the area less than 100 square meters is classified as a small pharmacy. OR=odds ratios; CI= confidence interval.

**Table 2 antibiotics-09-00163-t002:** Inquiries and counseling during community pharmacy visits in Shenyang.

Question Items	All Cases (*n* = 147)	Pediatric Case (*n* = 73)	Adult Case (*n* = 74)	χ^2^	*p*-Value
	n (%)				
**Inquiries about the symptom**	**121 (82.3)**	**60 (82.2)**	**61 (82.4)**	**0.021**	**0.885**
Asked about the length of cough	24 (16.3)	13 (17.8)	11 (14.9)		
Asked about cough duration	1 (0.7)	0 (0.0)	1 (1.3)		
Asked about whether had sputum or not	113 (76.9)	54 (74.0)	59 (79.7)		
Asked about the color of sputum	15 (10.2)	5 (6.9)	10 (13.5)		
Asked about other accompanying symptoms (sore throat, runny nose)	49 (33.3)	27 (37.0)	22 (29.7)		
Other question (asking the cause or severity of cough)	32 (21.8)	17 (23.3)	15 (20.3)		
Other question (asking about specific symptoms)	14 (9.5)	4 (5.5)	10 (13.5)		
**Inquiries about previous treatment(s)**	**24 (16.3)**	**16 (21.9)**	**8 (10.8)**	**3.319**	**0.069**
Asked whether had taken any other medicine or not	22 (15.0)	14 (19.2)	8 (10.8)		
Asked whether had seen a doctor or not	2 (2.7)	2 (2.7)	0 (0.0)		
**Inquiries about drug allergy and medical history**	**57 (38.8)**	**31 (42.5)**	**26 (35.1)**	**0.832**	**0.362**
Asked history of drug allergy	52 (35.4)	29 (39.7)	23 (31.1)		
Asked other medical history	5 (3.4)	2 (2.7)	3 (4.0)		
**Medication advice and other recommendations** *	**56 (38.1)**	**21 (28.8)**	**35 (47.3)**	**5.039**	**0.025**
Introduced the usage of the dispensed medicines	46 (31.3)	16 (21.9)	30 (40.5)		
Introduced the side effects of the dispensed medicines	5 (3.4)	2 (2.7)	3 (4.0)		
Others (advice on diet or other lifestyle factors)	6 (4.1)	1 (1.4)	5 (6.8)		
Recommended a referral to healthcare facilities	4 (2.7)	4 (5.5)	0 (0.0)		

Note: χ^2^ tests were used to compare the frequency of question-asked between pediatric and adult cases. * where there was a statistically significant difference between the two cases. Items in bold are the conclusions of the major inquiry and counseling (including four part: inquiry of symptom, inquiry of previous treatment, inquiry of medical and allergy history, medication advice and other recommendations), and items not in bold are the specific questions or advice of each part.

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
