# Peer review of "Dispensing Antibiotics without a Prescription for Acute Cough Associated with Common Cold at Community Pharmacies in Shenyang, Northeastern China: A Cross-Sectional Study"

_antibiotics, 2020, doi:10.3390/antibiotics9040163_

Round 1
Reviewer 1 Report
dear author, it is excellent topic but some re-arrangement of the paper is required.
- some information regarding the regulatory issues in dispensing antibiotics in china is needed, i.e. is it legal to supply antibiotics without prescription written by qualified prescriber? and if the same pharmacist is allowed to prescribe and dispense.
- need to state regulations regarding supply of medications for paediatrics use in the introduction
- Mystery shopper or standardized patient/client are more commonly used and understood globally, please consider using global terminolgy
- i found it confusing mixing the paediatric and adult cases for the entire paper, please consider re-arranging all results from paediatric together, all adults results together and then compare them.
- were the client-pharmacist encounter recorded, if yes were they verified by the pharmacist after?
- are all those references used and saying new things?
thank you
Author Response
Dear Professor:
Thank you very much for the constructive and helpful comments on our submission. We have revised the manuscript accordingly and please find the revisions in the resubmission and responses to the comments in attachment. Please see the attachment. We hope that we have dealt satisfactorily with the comments and concerns.
We look forward to your further reviews.
Best regards,

Reviewer 2 Report
Manuscript title: Dispensing antibiotics without a prescription for acute cough associated with common cold at community pharmacies: a simulated client study.
Summary
Thanks for your efforts and the opportunity to review this manuscript. It is a very interesting study that reported antibiotic prescribing with and without a prescription for simulated patients with acute cough in Shenyang, north-eastern China. It is important because antibiotic overuse is a huge problem, specifically in countries where antibiotics are being prescribed without a medical prescription.
Here are a few concerns that I have with the manuscript:
Major
- It is a bit unclear for me what is the study design used to compare pharmacy practices. A randomised trial seems an applicable study design, where pharmacies are randomised to adult vs children simulated patients. Was this the study design used? The authors report using random numbers to select the sampled sub-district but only used convenience sampling to select pharmacies. It would be helpful to clarify the study design and report it as well in the study title.
- The authors report that Macrolides treatment is recommended for a respiratory tract infection. They have cited an out-of-date reference to a guideline that was updated in 2018 ‘as far as I know’ (doi: 10.21037/jtd.2018.09.153). Moreover, in Cochrane systematic reviews, antibiotics for acute cough are not recommended let alone Macrolide class antibiotics. The harms and benefits of using antibiotics for acute respiratory infections are now preferred to be communicated to patients using decision aids. Please revise this section of the Discussion and address it accordingly.
- The authors report the customer’s pressure as a factor that influenced antibiotic prescribing. However, it is often reported in other studies that this is mainly perceived by clinicians and what patients need is reassurance that their infection is not severe, and they do not require antibiotic treatment.
- It would be helpful to provide more background information on how patients are paying for antibiotic treatments. Is it out of pocket expenses or they are reimbursed by their insurance? Is it expensive, or similar price like other over-the-counter treatments? This may shed some light on the behaviour of prescribing or requesting an antibiotic treatment
- Please report if the funding body is involved in your study or not and what is their level of involvement?
- The authors did not have a clear statement on what future research in this area should focus on. Do they recommend targeting pharmacists with behaviour change interventions for reducing antibiotic prescribing over the counter? Or are they recommending targeting the issue at a policy level? Or further investigations are still needed? The authors reported similar issues in other districts in China, maybe it is time to introduce a national antimicrobial stewardship approach.
Minor
- Page 2, line 51: it would be helpful to provide data (if available) on how big the issue of antibiotic prescribing is, for medical and non-medical use of antibiotics in China.
- Page 2, line 68: Typo ‘Megacity’ instead of ‘Magacity’
- In the procedures, it would be helpful if ethics approval was obtained at the time of piloting visits or not. Were the pharmacy staff aware of the identity of the simulated client or consented to be approached?
- Was the identity of the simulated patient revealed at the end of the visit? How pharmacies consented to participate in the study? Were they approached by a different researcher?
- I am not sure that reporting the pharmacy size in table 1 provides additional relevant information. It is not clear as well the reported socioeconomic is for the pharmacy or for the district where the pharmacy is located
- The question items were collected by memorising the information. There could be a limitation to be reported here ‘recall bias’. Was the simulated patient able to recall all the information from all the pharmacies?
- Please clarify what are the levels of demands in figure 1.
- It would be helpful to add figure 2 as a supplement and create a new figure only showing antibiotic classes for adult vs paediatric cases
Recommendations:
Major revisions and re-submit
Author Response
Dear Professor:
Thank you very much for the constructive and helpful comments on our submission. We have revised the manuscript accordingly and please find the revisions in the resubmission and responses to the comments in attachment. Please see in the attachment. We hope that we have dealt satisfactorily with the comments and concerns.
We look forward to your further reviews.
Best regards,

Round 2
Reviewer 2 Report
Thank you for submitting your revised manuscript. The authors have responded to my comments, and made huge improvements. I would recommend the article for publication after the following minor change:
Minor change:
1- Page 9, line 225: The evidence shows that antibiotic treatment may shorten the duration of acute cough for about 16 hours. https://doi.org/10.1002/14651858.CD000245.pub4
Thanks for your efforts